# Genetic Characterization of the Acidic and Neutral Glycosphingolipid Biosynthetic Pathways in *Neurospora crassa*

**DOI:** 10.3390/microorganisms11082093

**Published:** 2023-08-16

**Authors:** Jannatul F. Shoma, Ben Ernan, Griffin Keiser, Christian Heiss, Parastoo Azadi, Stephen J. Free

**Affiliations:** 1Department of Biological Sciences, SUNY University at Buffalo, Cooke Hall Room 109, Buffalo, NY 14260, USA; 2Complex Carbohydrate Research Center, University of Georgia, Athens, GA 30602, USAazadi@ccrc.uga.edu (P.A.)

**Keywords:** acidic glycosphingolipid, neutral glycosphingolipid, glucosylceramide, galactosylceramide, fungal growth and development, neurospora, ceramide synthesis

## Abstract

Fungal glycosphingolipids (GSLs) are important membrane components which play a key role in vesicle trafficking. To assess the importance of GSLs in the fungal life cycle, we performed a mutant phenotypic study of the acidic and neutral GSL biosynthetic pathways in *Neurospora crassa*. GSL biosynthesis begins with two reactions leading up to the formation of dihydrosphingosine. The first of these reactions is catalyzed by serine palmitoyltransferase and generates 3-keto dihydrosphinganine. In *N. crassa*, this reaction is catalyzed by GSL-1 and GSL-2 and is required for viability. The second reaction is carried out by GSL-3, a 3-keto dihydrosphinoganine reductase to generate dihydrosphingosine, which is used for the synthesis of neutral and acidic GSLs. We found that deletion mutations in the acidic GSL pathway leading up to the formation of mannosylinositol-phosphoceramide are lethal, indicating that acidic GSLs are essential for viability in *N. crassa*. Once mannosylinositol-phosphoceramide is made, it is further modified by GSL-5, an inositol-phosphoceramide-B C26 hydroxylase, which adds a hydroxyl group to the amide-linked fatty acid. GSL-5 is not required for viability but gives a clear mutant phenotype affecting all stages of the life cycle. Our results show that the synthesis of mannosylinositol-phosphoceramide is required for viability and that the modification of the amide-linked fatty acid is important for acidic GSL functionality. We also examined the neutral GSL biosynthetic pathway and identified the presence of glucosylceramide. The deletion of neutral GSL biosynthetic genes affected hyphal morphology, vegetative growth rate, conidiation, and female development. Our results indicate that the synthesis of neutral GSLs is essential for normal growth and development of *N. crassa*.

## 1. Introduction

Fungal glycosphingolipids (GSLs) are clustered with sterols to form lipid rafts in the endoplasmic reticulum, Golgi apparatus, and sterol-rich plasma membrane domains. They play crucial roles in vesicular trafficking, cell polarization, hyphal growth, and fungal fitness [1,2,3,4,5]. The chemical structures of fungal GSLs and some of the enzymes involved in their synthesis are distinct from those in mammals [6]. Therefore, the enzymes of the fungal GSL biosynthetic pathways are ideal drug targets for the development of antifungal agents. This has resulted in an increased interest in characterizing them in greater detail [7,8,9,10].

GSLs have been isolated and identified from various fungi, including *Saccharomyces cerevisiae*, *Candida albicans*, *Fusarium graminearum*, *Neurospora crassa*, and *Aspergillus* spp., [11,12,13,14]. The basic structure of GSLs consists of a serine residue with a 16-carbon fatty acid attached to its COOH end and either a C16–18 fatty acid (for neutral GSLs) or a C18-26 fatty acid (for acidic GSLs) attached to its NH_3_ end. There are two pathways for glycosphingolipid synthesis in fungi, an acidic pathway leading from dihydrosphingosine to the formation of mannosylinositol-phosphoceramide and a neutral pathway leading from dihydrosphingosine to the formation of glucosylceramide and galactosylceramide (Figure 1). Dihydrosphingosine synthesis begins with the condensation of palmitoyl-CoA with serine by the enzyme serine palmitoyltransferase to form 3-keto dihydrosphinganine [15,16,17]. Serine palmitoyltransferase has two subunits encoded by *gsl-1* and *gsl-2*. The 3-keto dihydrosphinganine is then converted to dihydrosphingosine by 3-keto dihydrosphinganine reductase [14,18].

The fungal acidic GSL pathway begins with the hydroxylation of the attached palmotyl moiety by sphingolipid C4 hydroxylase, an enzyme that is present in fungi and plants, to generate phytosphingosine [19,20]. In *N. crassa*, this enzyme is encoded by *gsl-6*. The second step in the acidic GSL pathway is the addition of a C18-C26 fatty acid in an amide linkage to phytosphingosine and is carried out by phytoceramide synthase to create phytoceramide [21,22]. The *N. crassa*, *gsl-4* encodes phytoceramide synthase. In the third step of the acidic GSL pathway, inositol phosphate is transferred from phosphatidylinositol to phytoceramide by inositol phosphoceramide synthase to generate inositol-phosphoceramide [23]. The *gsl-11* gene encodes this enzyme in *N. crassa*. In filamentous fungi, inositol-phosphoceramide is further modified by the addition of a mannose residue to the inositol ring by the enzyme mannose inositol-phosphoceramide transferase as the fourth step in the acidic GSL biosynthetic pathway [24]. Mannose inositol-phosphoceramide transferase is encoded by *gsl-12* in *N. crassa*. Mannosylinositol-phosphoceramide is then further modified by the enzyme inositol-phosphoceramide-B C26 hydroxylase to add a hydroxyl group to the penultimate C bond in the amide-linked fatty acid [20,25]. This enzyme is encoded by the *gsl-5* gene in *N. crassa*.

The neutral GSL pathway also begins with dihydrosphingosine (Figure 1). Analyses of the *N. crassa* neutral GSL pathway have been previously reported on by Park et al., who identified the pathway intermediates [12], and by Huber et al., who further characterized pathway intermediates and showed that the pathway was important for vegetative growth and conidiation [26]. The first step of the pathway is the addition of a C16-18 fatty acid in an amide linkage to dihydrosphingosine and is catalyzed by ceramide synthase to produce dihydroceramide (DhCer) [21,22,27]. In *N. crassa*, *gsl-20* encodes the ceramide synthase enzyme. In the second step of the neutral GSL pathway, the palmityl fatty acid is then reduced by sphingolipid ∆4-desaturase to generate OH-∆4-ceramide [28]. The sphingolipid Δ4-desaturase is encoded by the *dcd* gene in *N. crassa*. In the third step, a double bond at C8–9 is generated in the palmatidyl fatty acid by sphingolipid 18-desaturase [29,30]. The *gsl-8* gene encodes the desaturase enzyme in *N. crassa*. The fourth step in the pathway is the introduction of a methyl group at C9 of the palmatidyl fatty acid by sphingolipid C9-methyltransferase, resulting in the formation of OH-∆4-∆8-9-methyl-ceramide [29,30,31,32,33]. This methyltransferase is encoded by the *N. crassa gsl-9* gene. The *gsl-9* gene has been shown to be identical to the *sc* (*scumbo*) gene, identified by Barratt et al. [34,35]. Mutants of *scumbo* were characterized as having a slow-spreading growth, knobby protrusions, and abnormal conidiation phenotype. The final steps of the neutral GSL biosynthetic pathway consists of the transfer of a sugar residue from UDP-glucose or UDP-galactose to the OH-∆4-∆8-9-methyl-ceramide [22,36,37,38]. The enzyme catalyzing the formation of galactosylceramide is unknown. The Neurospora glucosylceramide synthase is encoded by the *gsl-10*.

Although different steps in the acidic and neutral GSL pathways have been identified and characterized in different fungi, we wanted to carefully characterize their role in fungal growth and morphogenesis by looking at the contribution of each step of the pathway in *Neurospora crassa*. Many of the steps in the *N. crassa* neutral biosynthetic pathway were previously examined by Huber et al. and the neutral pathway was found to be important for vegetative growth and conidiation [26]. We expanded on these previous reports by identifying all the genes encoding the acidic and neutral GSL pathway enzymes and examining the phenotypes of knockout mutants during all phases of the Neurospora life cycle. Our results showed that mutations in serine palmitoyltransferase, which block the synthesis of both acidic and neutral sphingolipids, are lethal. We found that mutations affecting all of steps in the synthesis of the acidic GSL mannosylinositol-phosphoceramide are lethal, highlighting the importance of acidic sphingolipids for *N. crassa*. Mutants affected in the addition of a hydroxyl group to the amide-linked fatty acid of acidic GSLs are viable but affected in all stages of the life cycle. Mutations in the neutral GSL pathway are viable, but they are affected in all stages of the life cycle, indicating that neutral GSLs are important. The neutral sphingolipids from *N. crassa* have been previously examined by Lester et al., Park et al., and Huber et al. [11,12,26]. We analyzed the neutral GSL from wildtype and mutant isolates with a TLC assay to demonstrate that the wildtype makes glucosylceramide and a second glycoceramide, which we presume to be galactosylceramide based on the work of Lester et al., who demonstrated the neutral GSLs from *N. crassa* contained glucose and galactose [11]. Using TLC analysis, we show that the ∆*gsl-20* (ceramide synthase) and ∆*gsl-10* (glycosylceramide synthases) mutants were affected in the synthesis of neutral GSLs. Mutation of the glucosylceramide synthase (∆*gsl-10*), which affects the synthesis of glucosylceramide but not galactosylceramide, gives a less severe phenotype than mutations further upstream in the neutral GSL pathway. This suggests that galactosylceramide partially compensates for the loss of glucosylceramide but is not able to completely replace the functions provided by glucosylceramide. Collectively, our results emphasize the importance of acidic and neutral GSLs for the growth and development in *N. crassa*. 

## 2. Materials and Methods

### 2.1. Strains and Culture Conditions

The deletion mutants for the genes encoding GSL biosynthesis pathway enzymes and wildtype strains were taken from the Neurospora single gene deletion library, which was obtained from the Fungal Genetics Stock Center. The *N. crassa* deletion library is found in 96-well plates, each well of which containing a conidial suspension of a deletion mutant. The gene deletion mutants were generated as part of the Neurospora genome project by replacing gene coding regions with a hygromycin resistance cassette [39,40]. The strains were maintained on Vogel’s medium supplemented with 2% sucrose. Hygromycin at a concentration of 200 µg/mL was added to the medium for the heterokaryotic isolates (strains containing a mixture of wildtype nuclei and deletion mutant nuclei) [41]. The Δ*gsl-1 mata* (NCU06870), Δ*gsl-2 mata* (NCU00447) Δ*gsl-6 mata* (NCU06465), Δ*gsl-4 mata* (NCU00008), Δ*gsl-11 mata* (NCU02882), Δ*gsl-12 mata* (NCU07761), and Δ*gsl-9 mata* (NCU07859) deletion mutants are found as heterokaryotic isolates in the Neurospora knockout library. The library contains homokaryotic isolates (strains in which all the nuclei are deletion mutant nuclei) Δ*gsl-3 matA* (NCU00302), Δ*gsl-5 matA* (NCU03492), Δ*gsl-20 mata* and Δ*gsl-20 matA* (NCU02468), Δ*dcd mata* and Δ*dcd matA* (NCU08927), Δ*gsl-8 mata* (NCU02408), Δ*gsl-10 mata*, and Δ*gsl-10 matA* (NCU01116). Standard mating experiments were used in an effort to get homokaryotic deletion mutant isolates for those mutants that were found as heterokaryons in the library and to obtain isolates of both mating types for those homokaryotic mutants in which the library contained a single mating type isolate [41]. In these experiments, as well as in mating experiments with the other mutants, the knockout mutations co-segregated with the mutant phenotypes, demonstrating that the mutations were responsible for the mutant phenotypes. We also used conidial streaking experiments in our efforts to obtain homokaryotic isolates from the heterokaryotic mutants [41]. 

### 2.2. Vegetative Growth Rate Assay

To determine the linear growth rate of the mutant strains, 5 µL of conidia from a mutant isolate (5 × 10^4^ conidia in water) was spotted near the edge of a Petri dish containing Vogel’s 2% sucrose 2% agar medium and the mutants were grown at 30 °C. The extension of the hyphae across the agar medium was monitored by marking the location of the leading edge of the colony at 10 h and 20 h post-inoculation. The linear growth rate was calculated as the average hourly rate of extension of the colony leading edge in the time interval. These linear growth rate experiments were performed in triplicate and an average growth rate with a standard deviation was determined for the wildtype and mutant strains. To examine hyphal morphology, the growing edges of the colonies were viewed in a dissecting microscope and photographed with a Canon Powershot A620 camera fitted with a microscope adaptor.

### 2.3. Conidia Production Assay

To assess how loss of the enzymes in the neutral GSL pathway affect conidia (asexual spore) production, we inoculated slant tubes containing 3 mL of Vogel’s sucrose medium with wildtype and mutant strains and allowed the strains to grow at 30 °C for 10 days and produce conidia. The conidia were harvested in water and the number of conidia produced was determined using a hemocytometer. These conidia production experiments were performed in triplicate and an average conidia/slant with a standard deviation was calculated. 

### 2.4. Ascospores Production Test

To characterize female development, mating experiments were carried out in triplicate between isolates of opposite mating types lacking the same step in the sphingolipid pathway. Conidia from the isolates were inoculated on two sterile 6 cm in diameter discs of Whatman 3 MM chromatography paper placed in a Petri dish with 5 mL of sterile synthetic crossing medium [41]. Whatman 3 MM paper serves as a carbon and energy source for the fungus. The strains were allowed to grow in the dark at room temperature for eighteen days. During this time, the fungus produced perithecia, the *N. crassa* female mating structure. Sterile water was added as needed to maintain a moist state in the filters. By eighteen days post-inoculation, wildtype perithecia had generated mature ascospores that were ejected onto the lid of the Petri dish. The ascospores were collected from the lid in sterile water using Pasteur pipettes and the number of ascospores was counted using a hemocytometer. Perithecia were manually collected and squashed between a microscope slide and a cover glass to break the perithecia open and release “rosettes” of developing ascospores. A Brightfield microscope was used to view the developing ascospores, which were photographed with a Canon Powershot A620 camera fitted with a microscope adaptor.

### 2.5. Characterization of Neutral GSLs

To examine the neutral sphingolipids present in deletion mutants in the neutral sphingolipid pathway, wildtype, Δ*gsl-20* (ceramide synthase), and Δ*gsl-10* (glucosylceramide synthase) mutant isolates were grown in 100 mL of Vogel’s sucrose liquid medium at room temperature in an orbital shaker for between 48 and 72 h. The mycelium was collected on a Buchner funnel and the neutral sphingolipids isolated using the protocol described by Tani et al. [13]. Briefly, the mycelium was ground to a fine powder in a mortar and pestle under liquid nitrogen and the sample was extracted with 10 mL of chloroform/methanol/water (2:1:0.8 by volume) at 37 °C for 16 h. The sample was then extracted a second time with chloroform/methanol/water (2:1:0.8) at 50 °C for 2 h and the two organic phases were combined and dried. The dried sample was dissolved in 4 mL of 0.5 M KOH in methanol/water (95:5 by volume) and incubated at 37 °C for 16 h to subject the sample to a mild alkaline hydrolysis. The sample was acidified to pH 1 by the addition of concentrated HCL and the lipids were extracted by the addition of chloroform, methanol, and water to generate a chloroform/methanol/water (2:1:0.8) mixture (taking into account the methanol and water that are in the sample). This was followed by a second extract with chloroform/methanol/water (2:1:0.8). The two organic phases were combined, dried, and dissolved in 10 mL of chloroform/methanol/water (30:60:8 by volume). To remove acidic lipids, the sample was applied to a DEAE-Sephadex column (5 mL bed volume) that had been equilibrated with chloroform/methanol/water (30:60:8), and the neutral lipids were eluted with 50 mL of chloroform/methanol/water (30:60:8). The samples were then dried, dissolved in 100 µL of chloroform/methanol/water (30:60:8), and spotted onto TLC plates for analysis.

Thin-layer chromatography was carried out on TLC silica gel 60 plates (Merck, Darmstadt, Germany) that had been immersed in ethanol containing 1.8% boric acid for 15 min and dried at 80 °C for 60 min. Samples were dotted onto the TLC plate, dried under a heat lamp, and subjected to chromatography with a chloroform/methanol/water solvent (60:35:8 by volume) for 45 min [13]. The TLC plates were stained by spraying the plates with a 1% orcinol in ethanol/sulfuric acid (70:3 by volume) solution and heating the plates to 100 °C for 15 min [42]. Individual sphingolipids were obtained by subjecting the samples to TLC analysis and scraping off the silica containing the purified sphingolipids. 

### 2.6. GC-MS Analysis of the Sugars Present in Purified Sphingolipid Samples

An analysis of the sugars present in the purified sphingolipids was performed by combined gas chromatography-mass spectrometry (GC-MS) of o-trimethylsilyl (TMS) methyl glycoside derivatives produced from the TLC-purified ceramides by acidic methanolysis [43]. The GC-MS analysis was performed on an AT7890A GC interfaced to a 5975B MSD, using an EC-1 fused silica capillary column (30 m × 0.25 mm ID) and a temperature gradient. 

## 3. Results

### 3.1. Genetic Characterization of the Entry Steps in Fungal GSL Biosynthesis

The first steps in GSL biosynthesis are the formation of 3-keto dihydrosphinganine by serine palmatidyl transferase, carried out by GSL-1 and GSL-2, and its conversion to dihydrosphingosine by 3-keto-dihydrosphinganine reductase (GSL-3). The *gsl-1* and *gsl-2* deletion mutants were found as heterokaryotic isolates (isolates with a mixture of wildtype and mutant nuclei) in the Neurospora knockout library. Efforts to isolate homokaryotic mutants (isolates containing only mutant nuclei) by mating the heterokaryotic mutants with wildtype strains of the opposite mating type or by conidia streaking on sorbose agar plates [41], were unsuccessful. In the conidial streaking experiments, we did see a few germlings with very weak abnormal growth which did not continue to grow when transferred. We concluded that the ∆*gsl-1* and ∆*gsl-2* mutations are lethal and that GSLs are needed for viability. The Neurospora knockout library did contain a putative homokaryotic mutant for *gsl-3*, which encodes the 3-keto dihydrosphinganine reductase needed to generate the dihydrosphinosine used in the acidic and neutral GSL pathways. However, PCR analysis showed that the *gsl-3* gene was intact in this putative deletion mutant, so a deletion mutant for 3-keto dihydrosphinganine reductase was not included in our analyses. 

### 3.2. Genetic Characterization of the Acidic GSL Pathway

The deletion mutants for the four steps of the acidic GSL pathway (Figure 1) Δ*gsl-6*, Δ*gsl-4*, Δ*gsl-11*, *and* Δ*gsl-12* were all found as heterokaryotic isolates in the Neurospora deletion library. Efforts to obtain homokaryotic isolates by mating or by conidia streaking experiments were unsuccessful. We concluded that mannosylinositol-phosphoceramide is required for viability in *N. crassa*. 

Once mannosylinositol-phosphoceramide is formed, it is further modified by GSL-5, which encodes an inositol-phosphoceramide-B C26 hydroxylase enzyme that generates a hydroxyl group near the terminus of amide-linked fatty acid [20,25]. The Neurospora deletion library contains a single homokaryotic *gsl-5* deletion mutant of mating type A and we obtained a *gsl-5 mata* isolate by mating the original mutant with a wildtype mating type A strain. We then characterized the *gsl-5* mutants and found that they were affected in all aspects of the Neurospora life cycle. We found that Δ*gsl-5* mutants have a hyperbranching vegetative morphology (Appendix A), a 79 +/− 3% reduction in their linear growth rate, and an 82 +/− 2% reduction in the production of conidia. The Δ*gsl-5* mutants were also affected in female development. Δ*gsl-5* × Δ*gsl-5* matings generated abnormal perithecia and no ascospores were produced (Appendix A). From these results, we conclude that the modification of the amide-linked fatty acid of mannose-inositol-phosphoceramide by inositol-phosphoceramide-B C26 hydroxylase is important for the functionality of the acidic GSL and required for ascospore development. 

### 3.3. Genetic Characterization of the N. crassa Neutral GSL Pathway

A BLAST search of the *N. crassa* genome for the neutral GSL pathway enzymes demonstrated that the genome contained the genes in the neutral GSL pathway (Figure 1**)**. The *N. crassa* genome contains a ceramide synthase gene (*gsl-20*), a sphingolipid ∆4-desaturase gene (*dcd*), a sphingolipid ∆8-desaturase gene (*gsl-8*), and a sphingolipid C9-methyltransferase gene (*gsl-9*). The glucosylceramide synthase gene (*gsl-10*) and an unidentified gene encoding a ceramide galactosyltransferase represent the final steps in the GSL biosynthesis. Although the genes responsible for glucosylceramide biosynthesis have been identified, cloned, and studied in several fungi [12,26,37,38], information on the galactosyltransferase responsible for the synthesis of galactosylceramide is lacking.

Homokaryotic mutant isolates were available for most of the steps in the neutral GSL pathway in the Neurospora knockout library. However, the Δ*gsl-9* mutant was provided as a heterokaryotic isolate, and homokaryotic isolates for Δ*gsl-8* and Δ*gsl-10* were only provided in one mating type. The ∆*gsl-9* heterokaryotic isolate was subjected to mating and conidia streaking to get homokaryotic ∆*gsl-9* strains, and both types of isolation procedures were successful. Mating experiments for Δ*gsl-8*, ∆*gsl-9*, and Δ*gsl-10* were successful in providing strains of both mating types for these deletions. We were thus able to get homokaryotic isolates for all the steps in the neutral GSL pathway and concluded that the neutral GSL pathway is not required for viability in *N. crassa*. 

To determine whether the neutral GSL biosynthesis pathway was important for hyphal morphology and vegetative growth, we carried out a phenotypic analysis of pathway mutants. The deletion mutants were examined for vegetative hyphae linear growth rate and for any morphological differences. We found that the mutants affected in the first three steps of the pathway, ∆*gsl-20*, ∆*dcd*, and ∆*gsl-8* showed significant morphological differences and produced shorter ariel hyphae than the wildtype control (Figure 2). The ∆*gsl-9* and Δ*gsl-10* mutant phenotypes were not quite as severe as that of the mutants affecting the earlier steps in the pathway (Figure 2). The mutant strains ∆*gsl-20*, ∆*dcd*, and ∆*gsl-8* showed significantly reduced linear growth rates (approximately 15% of the wildtype linear growth rate), demonstrating that neutral GSLs are important for the vegetative growth in *N. crassa* (Figure 3). The ∆*gsl-9* mutant (sphingolipid C9-methyltransferase mutant) was less severely affected than the ∆*gsl-20*, ∆*dcd*, and ∆*gsl-8* mutants. It had a growth rate of 59 +/− 5% of the wildtype growth rate, suggesting that some de-methylated glucosylceramide and de-methylated galactosylceramide are being synthesized from the OH-∆4-∆8 ceramide intermediate. The Δ*gsl-10* mutant, which is specifically affected in glucosylceramide synthesis, also had a less severe phenotype with 32 ± 3% of WT linear growth rate (Figure 3). Based on this observation, we conclude that mutants lacking glucosylceramide but able to make galactosylceramide showed less severe defects in morphology and vegetative growth than mutants earlier in the pathway, which lack both neutral sphingolipids.

We also looked at the ability of the neutral GSL pathway mutants to produce conidia (asexual spores) and found that conidia production was significantly reduced (Figure 4). When compared to a wildtype control isolate, the Δ*gsl-20*, Δ*dcd*, Δ*gsl-8*, and Δ*gsl-9* mutants showed a >99%, >99%, >99%, and 98% reduction in conidia production (Figure 4). The Δ*gsl-10* mutant showed a less severe phenotype than the upstream steps in the pathway, with a 93% reduction in conidia production demonstrating that for the conidiation phase of the life cycle, loss of glucosylceramide was less deleterious than the loss of both glucosylceramide and galactosylceramide.

To assess the importance of the neutral GSL pathway in perithecia and ascospore (sexual spore) development, mutant x mutant matings were performed for all the steps in the pathway. We found that in the Δ*gsl-20* × Δ*gsl-20*, matings protoperithecia-like structures were formed but they failed to continue development into perithecia (Figure 5). GSL-20 catalyzes the first step in the pathway, the conversion of dihydrosphingosine to a ceramide (Figure 1). The Δ*dcd* × Δ*dcd*, Δ*gsl-8* × Δ*gsl-8*, Δ*gsl-9* × Δ*gsl-9*, and Δ*gsl-10* × Δ*gsl-10* matings did produce perithecia, which tended to be smaller than the large healthy perithecia found in a wildtype × wildtype mating. In these mutant matings, we rarely saw ascospores being “shot” onto the lids of the Petri dishes in which the perithecia were being produced, while in control matings, large numbers of ascospores were shot onto the Petri dish lid. The numbers of ascospores produced in a Δ*gsl-10* × Δ*gsl-10* mating and a wildtype × wildtype mating were quantified, and the results showed the mutant mating had a 98% reduction in the number of ascospores produced. We performed perithecia squashes to further examine ascospore development in the ∆*gsl-10* × ∆*gsl-10* matings. The results showed that most of the perithecia were barren without any ascospores, but a few of the perithecia did produce a reduced number of ascospores. Figure 6 shows an example of one of the few *gsl-10* × *gsl-10* perithecia squashes in which we were able to see some developing ascospores. Many of these ascospores had a more rounded shape instead of the typical “American football” shape of wildtype ascospores, suggesting there were developmental difficulties in the development of the few ascospores that were produced. From our results, we conclude that the formation of a ceramide by GSL-20 is required for a protoperithecia to continue development into a perithecia and that the remaining steps in the pathway are important for perithecia and ascospore development.

### 3.4. Characterization of Sphingolipids with TLC

To examine the GSLs present in our samples and to verify that they were absent in mutant isolates, we purified and characterized the neutral GSLs from the wildtype isolate, the ∆*gsl-20* isolate (which lacks the neutral GSLs), and the ∆*gsl-10* isolate (which lacks glucosylceramide). The GSLs were purified and subjected to TLC analysis, as described in the Materials and Methods section. As shown by the arrows in Figure 7, the wildtype sample contains two GSLs that are not present in the ∆*gsl-20* sample, suggesting that these GSLs are the *N. crassa* glucosylceramide and galactosylceramide. The ceramide with slower mobility (shown with the broader arrow) is clearly lacking in the ∆*gsl-10* and ∆*gsl-20* samples, identifying this ceramide as glucosylceramide. The ceramide with faster mobility (shown with the narrow arrow) is lacking in the ∆*gsl-20* sample and there appears to be a very small amount of it in the ∆*gsl-10* sample, suggesting that it is galactosylceramide (Figure 7). In characterizing *N. crassa* GSLs, Lester et al. showed that the major sugars present in the GSLs were glucose and galactose [11]. In addition to the two presumptive neutral GSLs, the TLCs contained some additional bands that were not further characterized. In summary, we identified the GSLs present in our samples and verified that the ∆*gsl-20* mutant lacks neutral GSLs while the ∆*gsl-10* mutant lacks glucosylceramide and contains galactosylceramide. The TLC analysis also suggests that glucosylceramide is more abundant in the wildtype cell than galactosylceramide (Figure 7). While Huber et al. and Park et al. did excellent jobs in characterizing the sphingolipids in *N. crassa* by HPLC and mass spectrometry [12,26], their analyses did not discriminate between glucosylceramide and galactosylceramide.

### 3.5. Characterization of the Sphingolipids by GC-MS

To verify that the lower band in the TLC plate was glucosylceramide and to determine if the upper band was galactosylceramide, the purified sphingolipids were scraped off TLC plates and analyzed by gas chromatography and mass spectrometry. With the small amount of material available from the upper band, we were unable to identify the sugar(s) present in the sample. GC-MS analysis of the lower band showed that glucose was the most abundant sugar present with smaller amounts of mannose and xylose being detected in the analysis. We concluded that the lower band is glucosylceramide. This substantiates the conclusions we reached in the TLC analysis and further verifies the identification of GSL-10 as a glucosyltransferase.

## 4. Discussion

GSLs are involved in key biological processes, including cell signaling and vesicular transport of secretory proteins. GSLs have been found in lipid rafts and in sterol-rich plasma membrane domains [44,45]. GSLs in lipid rafts have been shown to facilitate the targeting of glycoproteins for secretion. Proteins with longer transmembrane region and GPI anchor proteins aggregate within these rafts and are transported through the secondary pathway. Hence, mutations in GSL biosynthesis could affect protein trafficking through the secretory pathway. We have carried out a genetic and phenotypic analysis of the acidic and neutral GSL biosynthetic pathways on the model filamentous fungus *N. crassa*. Both pathways begin with serine palmitoyltransferase and 3-keto dihydrosphingonanine reductase to generate dihydrosphingosine, which is used for the synthesis of both acidic and neutral GSLs (Figure 1). We determined that mutations in the *gsl-1* and *gsl-2* genes, which encode two subunits of serine palmitoyltransferase, are lethal and conclude that GSLs are required for viability in *N. crassa*. 

Our characterization of the acidic GSL pathway showed that the deletions of the four genes encoding the enzymes needed to synthesize mannosylinositol-phosphoceramide, *gsl-6, gsl-4*, *gsl-11*, and *gsl-12*, were lethal, and we concluded that mannosylinositol-phosphoceramide was needed for viability in *N. crassa*. This is in sharp contrast with the situations found in *S. cerevisiae*, *A. nidulans*, and *A. fumigatus*. In *S. cerevisiae*, the sphingolipid C4 hydroxylase gene (sur-2) is not essential for growth [20]. In *A. nidulans*, the loss of sphingolipid C4 hydroxylase or phytoceramide synthase generates a mutant phenotype with a reduction in growth rate and a hyperbranching vegetative hyphae [46]. Characterization of an *A. fumigatus* mutant affected in mannose inositol-phosphoceramide transferase, which catalyzes the formation of mannosylinositol-phosphoceramide from inositol-phosphoceramide, showed it to have a normal growth rate and morphology. This indicated that in *A. fumigatus*, inositol-phosphoceramide is adequate [24], while in *N. crassa*, loss of the enzyme is lethal.These comparisons of acidic GSL pathway enzymes in different fungi demonstrate that while the acidic GSL pathway is required for viability in *N. crassa*, this is not true for other characterized ascomycetes. We conclude that the importance of acidic GSLs varies from one fungus to another. We found that deletion in the *gsl-5* gene (inositolphosphoceramide-B C26 hydroxylase), which modifies mannosylinositol-phosphoceramide by placing a hydroxyl group in the penultimate position of the amide-linked fatty acid are viable but affected in all stages of the *N. crassa* life cycle. This indicates that the modification of the fatty acid is important for acidic GSLs. This hydroxylase (SCS7) is not essential for growth in *S. cerevisiae* [20,25]. 

The *N. crassa* neutral GSL biosynthetic pathway was previously studied and found to be needed for normal hyphal growth, hyphal morphology, and conidiation [26]. In their study, Huber et al. used HPLC and mass spectrometry to identify the neutral *N. crassa* sphingolipids [26]. Our work corroborates their findings and expands on their work by including sexual development in our phenotypic characterization and by examining the acidic GSL biosynthetic pathway. We found that mutations disrupting the earlier steps in neutral GSL synthesis (Δ*gsl-20*, Δ*dcd*, and Δ*gsl-8*) generate a severe mutant phenotype with a dramatically reduced linear growth rate, clear morphological changes, reduced conidiation, and disruption of female development (Figure 2, Figure 3 and Figure 4). The ∆*gsl-20*, ∆*dcd*, and ∆*gsl-8* mutants lack glucosylceramide and galactosylceramide. The ∆*gsl-9* mutant affected in the addition of a methyl group to the palmityl fatty acid is less severely affected in its growth rate than the other mutants, but is unable to make conidia; we suggest that the ∆*gsl-9* mutant may be able to make a small amount of glucosylceramide and galactosylceramide lacking the methyl group. In their analysis of the *N. crassa* neutral GSL pathway, Huber et al. provide some evidence for the presence of glycosylceramides in the ∆*gsl-9* mutant [26]. The ∆*gsl-10* mutant is less severely affected in both vegetative growth rate and in conidiation than the ∆*gsl-20*, ∆*dcd*, and ∆*gsl-8* mutants. In our TLC analysis, we showed that the ∆*gsl-10* mutant lacks glucosylceramide but does have a small amount of a second ceramide (presumably galactosylceramide) (Figure 7). The *gsl-10* mutant phenotype clearly shows that glucosylceramide is needed for normal growth, morphology, and female development (Figure 2 and Figure 3). The mutant phenotype of the *gsl-10* mutant indicates that the galactosylceramide being produced in the mutant is unable to fully compensate for the loss of glucosylceramide. However, the less severe phenotype of the ∆*gsl-10* mutant suggests that galactosylceramide also contributes to growth, morphology, and female development. In looking at the female developmental phenotypes associated with deletion mutants in the neutral GSL pathway, we found that mutants lacking GSL-20 (ceramide synthase) were unable to produce perithecia, while the deletion mutants in the later steps of the pathway produced small perithecia with a dramatically reduced number of ascospores. These results indicate that the production of a neutral ceramide is required for perithecia formation. The later steps of the neutral GSL pathway are needed for normal perithecia and ascospore development. 

Mutants affected in the synthesis of neutral GSLs have been studied in other fungi. *F. graminearum* and *A. nidulans* mutants in the ceramide synthase gene were found to be unable to generate perithecia and to be affected in morphology and growth rate [27,46], a phenotype we observed in our mutants (Figure 5). *C. neoformans* mutants affected in sphingolipid methyltransferase were found to have reduced virulence and to make de-methylated glucosylceramide [31]. In our analysis of the *N. crassa gsl-9* mutants, the phenotypic analysis suggested that these mutants might be making de-methylated glucosylceramide and de-methylated galactosylceramide. Sphingolipid methyltransferase mutants have also been studied in *A. nidulans*, *F. graminearum*, and *C. albicans* and were found to have defects in hyphal growth and conidia production [30,32,33]. 

In characterizing the neutral GSL pathway, we carried out TLC and a glycolipid carbohydrate composition analysis to demonstrate that the neutral GSLs were lacking in the ∆*gsl-20* mutant and that the ∆*gsl-10* mutant lacked glucosylceramide. In carrying out these analyses, we verified that ∆*gsl-20* and ∆*gsl-10* genes carried out the assigned enzymatic activities and showed that glucosylceramide was the more abundant neutral GSL in *N. crassa*. The gene encoding the enzyme(s) that participate in the conversion of OH-∆4-∆8-9 methylceramide to galactosylceramide remains to be elucidated. 

The GSL pathways have been considered good targets for the development of antifungal agents [7,8,9]. Our results suggest that anti-fungal agents directed against the acidic GSL pathway might be effective against plant pathogenic fungi closely related to *N. crassa*, but not against *A. fumigatus* and yeast pathogens. Developing anti-fungal agents targeting the enzymes of the neutral GSL pathway, particularly ceramide synthase (GSL-20) might be an effective strategy for the development of an antifungal that would be active against a broad group of fungal pathogens. 

## Figures and Tables

**Figure 1 microorganisms-11-02093-f001:**
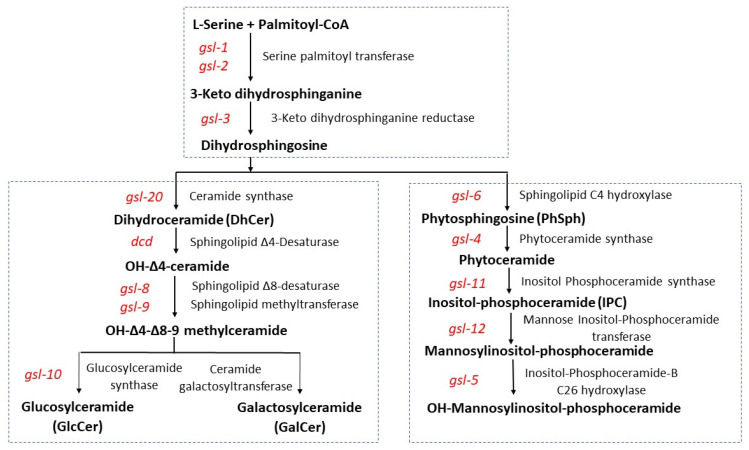
*Neurospora crassa* GSL biosynthetic pathways. The GSL pathway intermediates and the enzymes that catalyze the various reactions are noted. The *N. crassa* genes encoding the various steps in the pathways are given in italics.

**Figure 2 microorganisms-11-02093-f002:**
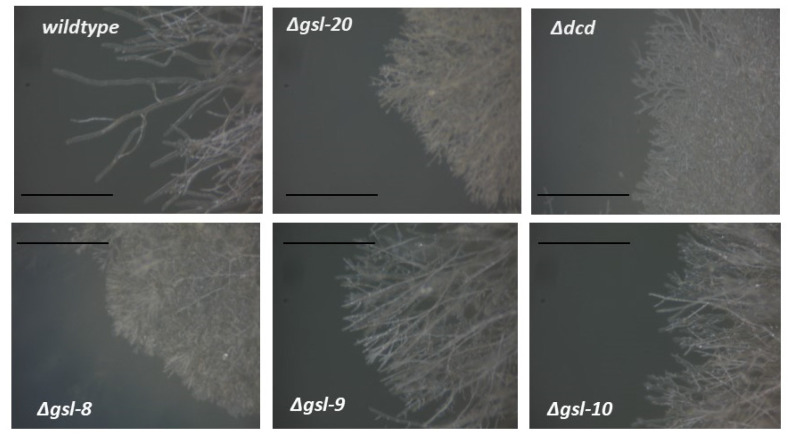
Morphology of the wildtype strain and mutant strains lacking neutral sphingolipids. Wildtype and mutant isolates were inoculated on Vogel’s 2% sucrose agar medium, and the growing edges of the colonies were viewed in a dissecting microscope. The edges of the growing colonies were photographed with a Canon Powershot A620 camera using overhead lighting. The scale bar shows 1 mm in length.

**Figure 3 microorganisms-11-02093-f003:**
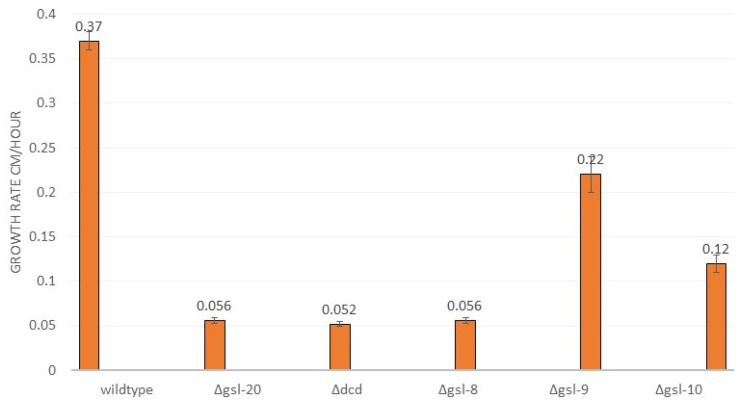
Comparison of linear hyphal growth rates of wildtype and mutant strains. Wildtype and mutant isolates were spotted near the edge of a Petri dish containing Vogel’s 2% sucrose agar medium and grown at 30 °C. The positions of the growing edges of the hyphal colonies were measured as a function of time to define a linear growth rate.

**Figure 4 microorganisms-11-02093-f004:**
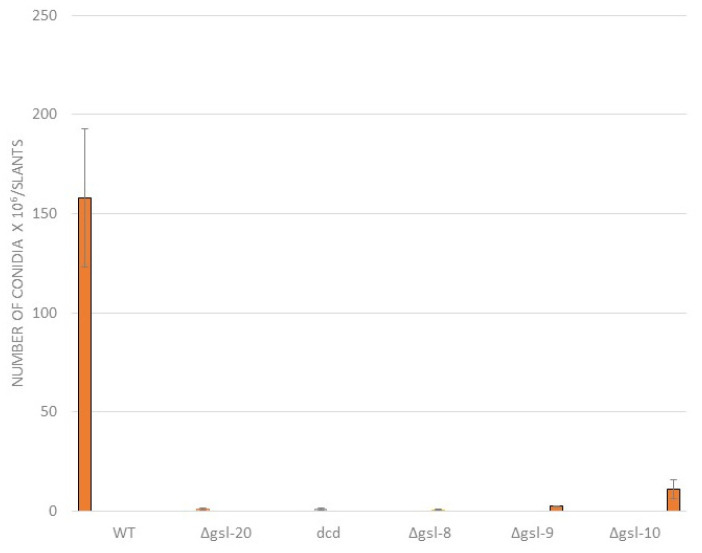
Comparison of conidia production by wildtype and mutant strains. Wildtype and mutant isolates were inoculated into agar slants of Vogel’s sucrose medium and allowed to grow and produce conidia for 10 days at 30 °C. The conidia were collected and the number of conidia produced was determined.

**Figure 5 microorganisms-11-02093-f005:**
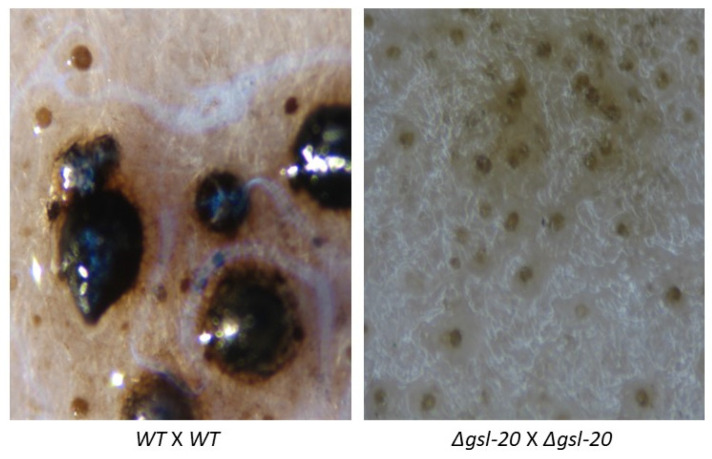
Perithecia production in wildtype and ∆*gsl-20* mutant mating experiments. Wildtype × wildtype and ∆*gsl-20* × ∆*gsl-20* matings were carried out by inoculating conidia on Whatman 3 MM paper with synthetic crossing medium and incubated at 22 °C for 14 days. Pictures of perithecia were taken with a Canon Powershot A620 camera with overhead lighting. Note the presence of large melanized perithecia in the WT × WT mating and the absence of perithecia in the ∆*gsl-20* × ∆*gsl-20* mating. The small round brownish structures in the ∆*gsl-20* × ∆*gsl-20* image are protoperithecia.

**Figure 6 microorganisms-11-02093-f006:**
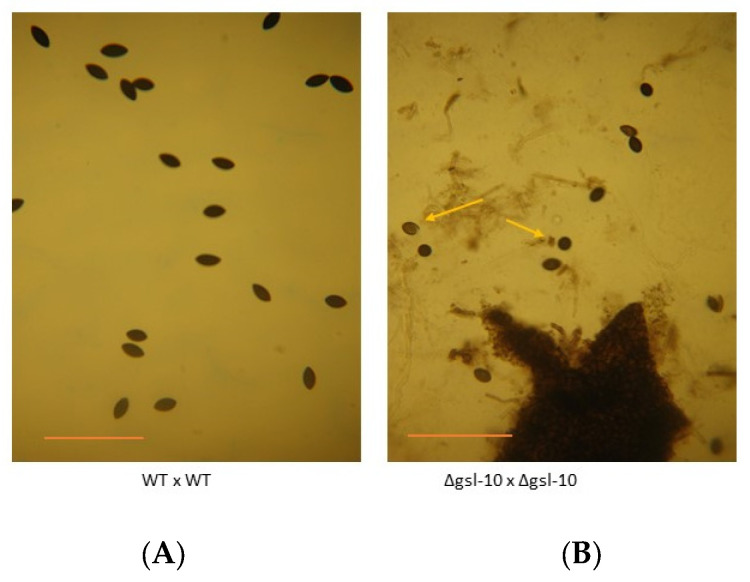
Ascospore development in wildtype and ∆*gsl-10* matings. Wildtype and ∆*gsl-10* matings were performed and the perithecia were allowed to develop for 18 days at 22 °C. Perithecia were collected and squashed to release ascospores. (**A**) In a wildtype × wildtype mating, healthy melanized ascospores were released from the perithecia. (**B**) Most perithecia in ∆*gsl-10* × ∆*gsl-10* matings were barren (no ascospores), but in approximately 5% of the perithecia, a few ascospores were produced. The dark object at the bottom of the ∆*gsl-10* × ∆*gsl-10* image is a perithecium fragment. Note that some of the ascospores have a rounded shape (see arrows) instead of the normal “American football” shape found in normal ascospores. The scale bar represents 100 µm.

**Figure 7 microorganisms-11-02093-f007:**
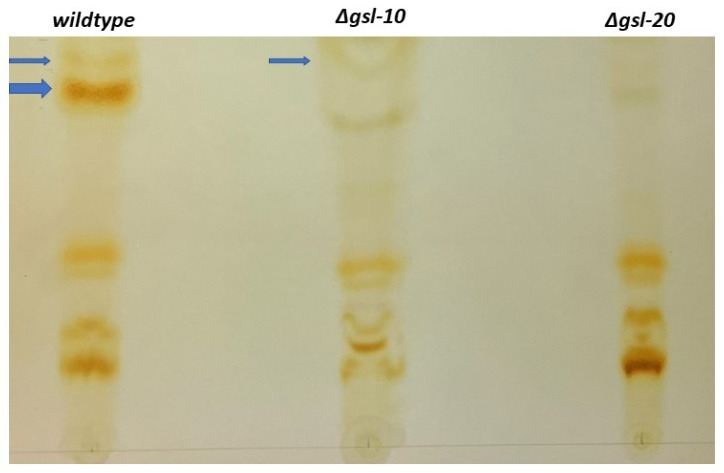
TLC analysis of neutral GSLs. GSLs were purified and subjected to chromatography on borate silica TLC plates. The thicker arrow points to the glucosylceramide in the GSLs from the wildtype isolate. The thinner arrows point to a second ceramide (presumably galactosylceramide) found in the wildtype and ∆*gsl-10* GSL samples. Note the absence of both ceramides in the ∆*gsl-20* sample.

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
