# Peer review of "Genetic Characterization of the Acidic and Neutral Glycosphingolipid Biosynthetic Pathways in Neurospora crassa"

_microorganisms, 2023, doi:10.3390/microorganisms11082093_

Round 1

Reviewer 1 Report

The article by Shoma et al, characterized both acidic and neutral Glycosphingolipid (GSL) in N. crassa. Acidic GSL are important for viability, while neutral GSL regulates cellular morphology, hyphal growth, and development. The paper is well written and the discussion is supported by strong datasets. Here are my comments:

1) Did the authors cross a WT with the mutants to see if the mutant phenotype can be reverted during the perithecia and ascospore development? Please clarify.

2) Can statistical analyses be done to bolster the datasets in Figs 2 and 3. Please clarify. If so statistical analyses should be showed in the Figs 2 and 3.

3)Why perithecia formation was only tested on gsl-20 mutant and not in the gsl-10 mutant. Please clarify.

Author Response

We have made some minor changes to our manuscript “Genetic Characterization of the Acidic and Neutral Glycosphingolipid biosynthetic pathways in Neurospora crassa” in response to the suggestions made by the reviewers.  I have uploaded the revised manuscript, with the changes highlighted in red.  The changes in the manuscript are detailed below. 

Reviewer #1 had three questions.

  • Did the authors cross a WT with the mutants to see if the mutant phenotype can be reverted during the perithecia and ascospore development? Please clarify.

In response to this query, we have inserted a new sentence in the materials and methods.  The new sentence reads “In these experiments, as well as in mating experiments with the other mutants, the knockout mutations co-segregated with the mutant phenotypes demonstrating that the mutations were responsible for the mutant phenotypes.” 

  • Can statistical analyses be done to bolster the data sets in Figs 2 and 3? Please clarify.  If so, statistical analyses should be shown in Figs 2 and 3.

We think the reviewer meant Figures 3 and 4 (Figure 2 is a morphology image while Figures 3 and 4 are growth rate and conidia production graphs where statistical data would be expected).   The data for these graphs was generated in triplicate experiments and both graphs contain standard deviation error bars.  In response to the question, we have added “These conidia production experiments were performed in triplicate and an average conidia/slant with a standard deviation was calculated.” to the Conidia production section of the Materials and Methods.   A similar statement is found in the vegetative growth rate assay section in Materials and Methods.

  • Why perithecial formation was only tested on gsl-20 mutants and not in the gsl-10 mutant. Please clarify.

The perithecia formation analyses included both gsl-20 and gsl-10.  In section 3.3 the manuscript addresses the gsl-20 mutants.  It reads “To assess the importance of the neutral GSL pathway in perithecia and ascospore (sexual spore) development, mutant x mutant matings were performed for all the steps in the pathway.  We found that in the Δgsl-20 x Δgsl-20 matings protoperithecia-like structures were formed but they failed to continue development into perithecia (Figure 5).”   A couple of lines further down we address the gsl-10 mutant along with the other steps in the neutral GSL pathway.   The manuscript reads “The Δdcd x Δdcd, Δgsl-8 x Δgsl-8, Δgsl-9 x Δgsl-9 and Δgsl-10 x Δgsl-10 matings did produce perithecia, which tended to be smaller than the large healthy perithecia found in a wildtype x wildtype mating.”

Reviewer 2 Report

Review for

Article

Genetic Characterization of the Acidic and Neutral Glycosphingolipid biosynthetic pathways in Neurospora crassa

by Jannatul F. Shoma1, Ben Ernan1, Griffin Keiser2, Christian Heiss2, Parastoo Azadi2 and Stephen J. Free1*

Fungal glycosphingolipids (GSLs) are important membrane components which play a key

role in vesicle trafficking. To assess the importance of GSLs in the fungal life cycle, authors performed

a mutant phenotypic study of the acidic and neutral GSL biosynthetic pathways in Neurospora crassa.

The paper covers a research field not so much covered in this organism.

Sphingolipid-enriched domains in fungi

Santos, F.C., Marquês, J.T., Bento-Oliveira, A., de Almeida, R.F.M.

FEBS Letters, 594(22), pp. 3698–3718, 2020

Membrane Sphingolipids Regulate the Fitness and Antifungal Protein Susceptibility of Neurospora crassa

Huber, A., Oemer, G., Malanovic, N., ...Keller, M.A., Marx, F.

Frontiers in Microbiology, 10(APR), 605, 2019

Changes in the biophysical properties of the cell membrane are involved in the response of Neurospora crassa to staurosporine

Santos, F.C., Lobo, G.M., Fernandes, A.S., Videira, A., De Almeida, R.F.M.

Frontiers in Physiology, 9(OCT), 1375, 2018

-----------------------

We conclude that the synthesis of mannosylinositol-phosphoceramide is required 22

for viability and that the modification of the amide-linked fatty acid is important for acidic GSL 23

functionality. We also examined the neutral GSL biosynthetic pathway and identified the presence 24

of glucosylceramide. We found that the deletion of neutral GSL biosynthetic genes affected hyphal 25

morphology, vegetative growth rate, conidiation, and female development. Our results indicate 26

that the synthesis of neutral GSLs is essential for normal growth and development of N. crassa. 27

I don’t like very much writing with so many  ‘We…’

---------------------------

Figures S1 and S2 not incorporated in main text? according to MDPI rules?

-----------------------

refs not in MDPI style

should be numbered

-----------------------

Author Response

We have made some minor changes to our manuscript “Genetic Characterization of the Acidic and Neutral Glycosphingolipid biosynthetic pathways in Neurospora crassa” in response to the suggestions made by the reviewers.  I have uploaded the revised manuscript, with the changes highlighted in red.  The changes in the manuscript are detailed below.  

Reviewer #2 indicated that the paper covers a research field not so much covered in this organism” and provided three references.   We noticed that two of these references were not included in our reference list, and so we have added them to the paper (references 44 and 45).

The review suggested that the manuscript should include fewer sentences beginning with “We” (particularly in the abstract which was quoted by the reviewer).  We have rewritten some sentences in response to this suggestion.  The abstract now reads “Our results show that the synthesis of mannosylinositol-phosphoceramide is required for viability and that the modification of the amide-linked fatty acid is important for acidic GSL functionality.  We also examined the neutral GSL biosynthetic pathway and identified the presence of glucosylceramide.  The deletion of neutral GSL biosynthetic genes affected hyphal morphology, vegetative growth rate, conidiation, and female development.  Our results indicate that the synthesis of neutral GSLs is essential for normal growth and development of N. crassa.”  In section 3.3 we also changed a “we performed” to “experiments were performed”.

Reviewer #2 pointed out that the supplemental figures were included in the manuscript and need to be moved to a supplemental figures file.   We have deleted Figures S1 and S2 from the manuscript and uploaded them as supplemental material.

Reviewer #2 pointed out that our references were not in the correct style.  We have changed the style to the MDPI reference style and thank the reviewer for pointing this out.